# Could Ovarian Cancer Prediction Models Improve the Triage of Symptomatic Women in Primary Care? A Modelling Study Using Routinely Collected Data

**DOI:** 10.3390/cancers13122886

**Published:** 2021-06-09

**Authors:** Garth Funston, Gary Abel, Emma J. Crosbie, Willie Hamilton, Fiona M. Walter

**Affiliations:** 1The Primary Care Unit, Department of Public Health and Primary Care, University of Cambridge, Cambridge CB1 8RN, UK; fmw22@medschl.cam.ac.uk; 2University of Exeter Medical School, University of Exeter, Exeter EX1 1TX, UK; g.a.abel@exeter.ac.uk (G.A.); W.Hamilton@exeter.ac.uk (W.H.); 3Gynaecological Oncology Research Group, Division of Cancer Sciences, University of Manchester, Manchester M13 9WL, UK; emma.crosbie@manchester.ac.uk; 4Manchester Academic Health Sciences Centre, Department of Obstetrics and Gynaecology, Manchester University NHS Foundation Trust, Manchester M13 9WL, UK; 5Institute of Population Health Sciences, Barts and The London School of Medicine and Dentistry, Queen Mary University of London, London E1 2AB, UK

**Keywords:** ovarian cancer, CA125, cancer antigen 125, primary care, diagnostic prediction model, early detection, risk assessment, triage, diagnostic pathways

## Abstract

**Simple Summary:**

Earlier detection of ovarian cancer has the potential to improve patient outcomes, including survival. However, determining which women presenting in primary care to refer for specialist assessment and investigation is a clinical dilemma. In this study, we used routinely collected English primary care data from 29,962 women with symptoms of possible ovarian cancer who were tested for the ovarian cancer biomarker CA125. We developed diagnostic prediction models to estimate the probability of the disease. A relatively simple model, consisting of age and CA125 level, performed well for the identification of ovarian cancer. Including additional risk factors within the model did not materially improve model performance. Following further validation, this model could be used to help triage symptomatic women in primary care based on their risk of undiagnosed ovarian cancer, identifying those at high risk for urgent specialist investigation and those at lower (but still elevated) risk for non-urgent investigation or monitoring.

**Abstract:**

CA125 is widely used as an initial investigation in women presenting with symptoms of possible ovarian cancer. We sought to develop CA125-based diagnostic prediction models and to explore potential implications of implementing model-based thresholds for further investigation in primary care. This retrospective cohort study used routinely collected primary care and cancer registry data from symptomatic, CA125-tested women in England (2011–2014). A total of 29,962 women were included, of whom 279 were diagnosed with ovarian cancer. Logistic regression was used to develop two models to estimate ovarian cancer probability: Model 1 consisted of age and CA125 level; Model 2 incorporated further risk factors. Model discrimination (AUC) was evaluated using 10-fold cross-validation. The sensitivity and specificity of various model risk thresholds (≥1% to ≥3%) were compared with that of the current CA125 cut-off (≥35 U/mL). Model 1 exhibited excellent discrimination (AUC: 0.94) on cross-validation. The inclusion of additional variables (Model 2) did not improve performance. At a risk threshold of ≥1%, Model 1 exhibited greater sensitivity (86.4% vs. 78.5%) but lower specificity (89.1% vs. 94.5%) than CA125 (≥35 U/mL). Applying the ≥1% model threshold to the cohort in place of the current CA125 cut-off, 1 in every 74 additional women identified had ovarian cancer. Following external validation, Model 1 could be used as part of a ‘risk-based triage’ system in which women at high risk of undiagnosed ovarian cancer are selected for urgent specialist investigation, while women at ‘low risk but not no risk’ are offered non-urgent investigation or interval CA125 re-testing. Such an approach has the potential to expedite ovarian cancer diagnosis, but further research is needed to evaluate the clinical impact and health–economic implications.

## 1. Introduction

In 2020, an estimated 313,959 women were diagnosed with ovarian cancer worldwide [1]. Survival is highly dependent on the stage at diagnosis, with five-year survivals of 93% and 13% for UK women diagnosed at stage I and IV, respectively [2]. Although several large screening trials have been conducted, they have failed to demonstrate a long term survival benefit [3,4]. In the absence of screening programs, the majority of women with ovarian cancer are diagnosed after they present with symptoms [5,6]; thus, timely diagnosis of these women may improve cancer outcomes.

In 2011, the National Institute for Health and Care Excellence (NICE) recommended that women presenting to their General Practitioner (GP) with symptoms of possible ovarian cancer in England, Wales, and Northern Ireland be tested for the serum biomarker cancer antigen 125 (CA125) [7]. Further investigation with ultrasound was advocated if CA125 levels were ≥35 U/mL. However, this threshold was not based on primary care evidence. In primary care prior to diagnosis, 23% of women with ovarian cancer have CA125 levels of less than 35 U/mL [8], and concerns have been raised over the potential clinical impact of delayed diagnoses in this false-negative group [9]. In addition, even if women have markedly elevated CA125 levels (indicating a very high likelihood of undiagnosed ovarian cancer), they still must undergo GP-requested ultrasound before they are eligible for an urgent (‘two-week wait’) cancer pathway referral to a specialist. 

We recently developed a prediction model based on age and CA125 level in order to estimate the probability of ovarian cancer in individual women undergoing testing in primary care in England [8]. If implemented in clinical practice, such models would allow women to be triaged using the risk of undiagnosed ovarian cancer so that those at greatest risk undergo urgent specialist investigation, those at ‘low risk but not no risk’ are monitored or undergo non-urgent investigation, and those at very low risk (the majority) can be reassured. Such primary care evidenced risk-based triage strategies may reduce diagnostic delay and thereby improve patient outcomes.

This study had three objectives. First, as previous studies have shown that a range of patient risk factors and blood tests can help predict undiagnosed ovarian cancer [10], we sought to develop a comprehensive diagnostic prediction model, making the best use of variables routinely available in primary care. Second, we aimed to compare the diagnostic performance of this model with that of a simpler model comprising age and CA125. Third, we sought to explore the potential implications of implementing different model risk thresholds on ovarian cancer detection in primary care.

## 2. Methods

This study is reported in accordance with transparent reporting of a multivariable prediction model for individual prognosis or diagnosis (TRIPOD) guidelines. A completed checklist is included (Appendix A).

### 2.1. Data Source

We conducted a retrospective cohort study using linked data from the Clinical Practice Research Datalink (CPRD) GOLD, Hospital Episode Statistics Admitted Patient Care dataset (HES APC), and the National Cancer Registration and Analysis Service (NCRAS). The CPRD GOLD dataset contains anonymised, coded, primary care data including demographics, laboratory results, and symptoms for around 11 million patients and is broadly representative of the UK population [11]. The HES APC dataset includes information about hospital admissions and patient demographics, including ethnicity [12]. The NCRAS (the English cancer registry) collects cancer registration data on patients, including detailed information on tumour topography and diagnosis date. The NCRAS obtains data from hospitals, GP surgeries, and death certificates and reports a near 100% case ascertainment [13]. Linkage of CPRD and NCRAS data was performed at a patient level by a third party, National Health Service (NHS) Digital [14]. The NCRAS only collects details of cancers diagnosed in England, so the study was restricted to English general practices. 

### 2.2. Participants

We identified women with a code for CA125 measurement in primary care between 1 May 2011 and 31 December 2014 and included all women who met the study criteria in order to maximise sample size. We excluded women who were <18 years old or registered at a GP practice not deemed ‘up-to-standard’ on data quality by CPRD on the date of their first CA125 test during this period [11]. Women with a record of ovarian cancer in NCRAS data on or before the CA125 test date were also excluded, as were women with a CA125 test in the 12 months before the first CA125 test during the study period. To maximise data quality, only CA125 entries recorded in standard equivalent units of CA125 measurement (U/mL, IU/mL, KU/L, KIU/L) were accepted, and CA125 values associated with spurious laboratory cut-offs (245, 420, and 455 U/mL), or those where no cut-off was given, were excluded. As our aim was to develop models for use in symptomatic women, and we wished to develop a model that included symptom variables, we restricted the cohort to those who had a recognised symptom of possible ovarian cancer recorded in the year prior to CA125 testing. A code list was used to identify symptoms included in current NICE guidelines on ovarian cancer detection [15]. Women with ascites or a palpable pelvic mass recorded prior to CA125 testing were excluded, as these clinical signs have high positive predictive values for ovarian cancer and warrant urgent referral to a specialist [15,16]. If women had more than one CA125 test during the study period, only the first was used in the analyses.

### 2.3. Outcome Definition

The primary outcome was a diagnosis of ovarian cancer, as recorded using International Classification of Disease (ICD)-10 codes in NCRAS data, in the 12 months following initial CA125 testing. It was assumed that cancer diagnosed within 12 months of the initial CA125 test was present at the time of CA125 testing. It is possible that incidental ovarian cancers might arise and be diagnosed in the 12 months after testing or that it may take more than 12 months following presentation for women to be diagnosed. We chose a period of 12 months, which has been applied widely in previous studies [8,17,18,19,20], as a compromise between minimising the inclusion of incidental ovarian cancers and maximising the inclusion of relevant cancers.

Ovarian cancer was defined as an ovarian malignancy (C56), a fallopian tube malignancy (C57.0), a peritoneal malignancy (C48.1, C48.2), or a neoplasm of uncertain behaviour of the ovary (D39.1) [21,22]. 

### 2.4. Prediction Models

#### 2.4.1. Candidate Variables

From the literature, we identified a list of candidate variables, which included (1) established ovarian cancer risk/protective factors, (2) ovarian cancer symptoms, and (3) blood tests. We focussed on variables that are well-recorded within GP records. While all patients in this study had a symptom of possible ovarian cancer prior to CA125 testing, previous research has demonstrated that these symptoms differ markedly in their predictive value [19]. Symptoms were therefore included as variables to account for this. Routine blood tests were included as recent studies have shown that both having a test performed and the specific test level can be predictive of undiagnosed cancer in primary care [17,18,23]. The final list of candidate variables taken forward into data-driven selection procedures was determined by the consensus of a multidisciplinary group consisting of GPs (GF, FMW, and WH), a gynaecological oncologist (EJC), and a statistician (GA) (Table 1). Ethnicity was classified into 5 categories (White, Mixed, Black, Asian, and Other) in line with the 2001 census, using a pre-developed code list [24]. These were subsequently collapsed into 2 groups—“White” and “Other ethnicities”—as numbers in individual ethnic groups other than White were small, and the risk of ovarian cancer is higher in people of White ethnicity than Black and Asian ethnicity [25]. Recent research indicates that patients who have had routine blood tests performed in primary care are at greater risk of undiagnosed cancer (even if the test results are normal) when compared with those who have not had routine blood tests performed [17]. Therefore, we included each routine blood test as a categorical variable, with “no test” forming a category. Full details of variable preparation are included in the Appendix A.

#### 2.4.2. Model Derivation and Internal Validation

We developed two models. Model 1 was pre-specified and contained age and CA125 level alone. Model 2 contained the most predictive variables selected from the list of candidates. 

Prior to model derivation, continuous variables were mean-centred. BMI and CA125 level were right-skewed, so they were log-transformed. The relationships between log CA125 and age with ovarian cancer were non-linear. To account for this, we generated restricted cubic splines (5 knots) for log CA125 and age and included these in place of age and log CA125 within the models [38]. The choice of 5 knots was pre-specified based on the results of previous research, which used the study data to explore the relationship between CA125 level, age, and ovarian cancer [8].

In order to derive Model 2, multivariate imputation by chained equation (MICE) [39] was used to replace missing data on ethnicity, height, and body mass index (BMI). Twenty imputations were performed. Following imputation, a logistic regression model containing all candidate variables was fitted to estimate variable coefficients. Rubin’s rules were used to combine the results across the imputed datasets [40]. Possible interaction between age and log CA125 level was examined through the inclusion of an interaction term. We examined 19 candidate variables with a total of 32 degrees of freedom (main effect and non-main effect), giving 9 ‘Events Per Parameter’ (EPP). To select variables for Model 2, a backward elimination approach was used in which the full model was fitted and the least significant variable was removed; then the model was refitted and the process repeated until all variables had a *p* value of ≤0.05. Variable coefficients were used as model weights.

To assess model discrimination, i.e., the ability of a model to distinguish those who have a disease from those who do not have a disease, we calculated the area under the receiver operating characteristic curve (AUC). Ten-fold cross-validation was performed to assess for any optimism in model discrimination. To examine model calibration, i.e., agreement between model estimated outcomes and observed outcomes, we calculated the calibration slope using 10-fold cross-validation. For Model 2, discrimination and calibration were calculated for each imputed dataset, and Rubin’s rules were used to combine results across the imputed datasets. 

#### 2.4.3. Model Thresholds

We devised thresholds to identify women with a ≥1%, ≥2%, and ≥3% probability of undiagnosed ovarian cancer based on our models. The lowest threshold, ≥1%, was selected, as patients have reported that they would opt for cancer investigations at this risk level [41]. The highest threshold, ≥3%, was chosen to match the ‘risk threshold’ at which NICE advocates urgent specialist cancer investigation or referral for symptomatic primary care patients [15]. We applied these thresholds to the study cohort to calculate their diagnostic accuracy (sensitivity, specificity, positive predictive value [PPV], and negative predictive value [NPV]) for the detection of ovarian cancer within 12 months of CA125 testing. We compared these accuracy metrics with that of CA125 at its standard, NICE-advocated cut-off (≥35U/mL) [7]. We also compared the accuracy of model thresholds to that of CA125 cut-offs with equivalent sensitivities to determine whether using the models, rather than just CA125-based cut-offs, improved accuracy. 

All analyses were performed in Stata version 15.1. The user-written cvAUROC command was used to calculate cross-validation AUCs [42].

## 3. Results

A total of 29,962 women met the inclusion criteria (Figure 1). Of these, 279 (0.9%) were diagnosed with ovarian cancer in the 12 months following CA125 testing.

The baseline characteristics of the cohort are shown in Table 2. Missing data were noted for ethnicity (*n* = 1234, 4.1%), height (*n* = 1721, 5.7%), and BMI (*n* = 2986, 10%). The mean patient age was 55 years (range 18–101 years). The most common symptom was abdominal/pelvic pain, recorded for 58.5% of women. The proportion of women who had blood tests recorded (other than CA125) ranged from 66% for CRP to 94.5% for haemoglobin.

### 3.1. Predictor Variables

The variables retained in Model 2 after backward elimination procedures were age, ethnicity, BMI, height, abdominal/pelvic pain, distension, CA125 level, platelet level, and albumin level. Coefficients and odds ratios for all variables are included in the Appendix A.

### 3.2. Model Discrimination and Calibration

The AUCs of Model 1 and 2, when directly calculated from the dataset (apparent performance), were similar (Table 3). On cross-validation, the models had the same AUC (0.935). There was little difference between apparent and cross-validation AUCs, indicating that model overfitting/optimism was minimal. The AUC of CA125 alone, calculated directly from the study cohort, was 0.932. Models 1 and 2 had calibration slopes close to 1, indicating good calibration, but confidence intervals were wide.

### 3.3. Thresholds for Further Investigation

As the more parsimonious Model 1 exhibited the same cross-validation AUC and similar calibration metrics to Model 2, the evaluation of thresholds for further investigation focussed on Model 1. The diagnostic accuracies of Model 1 thresholds for the detection of ovarian cancer within 12 months of CA125 testing are shown in Table 4. These were compared against CA125 cut-offs with equivalent sensitives for ovarian cancer. At the ≥1% probability threshold, the specificity of Model 1 was 3.1% higher than a CA125 cut-off with the same sensitivity (≥23 U/mL), while there was a less marked difference at higher model probability thresholds. At all model thresholds, the PPV was higher than that of CA125 cut-offs with equivalent sensitivities. 

The potential implications of applying different model thresholds to the study cohort of 29,962 women are illustrated in Figure 2. In comparison with the current CA125 cut-off, applying a probability threshold of ≥1% would result in an additional 1622 women being identified for further evaluation for ovarian cancer, of whom 22 (1.4%) would have ovarian cancer; i.e., an additional 1 in every 74 women identified for further evaluation would have ovarian cancer. Applying a ≥3% model probability threshold instead of the current CA125 cut-off would result in 706 fewer women being identified for further evaluation, of whom 8 (1.1%) would have ovarian cancer. Applying a 2% model threshold instead of the current CA125 threshold would result in 58 fewer women being identified for further evaluation, of whom none would have ovarian cancer.

## 4. Discussion

A model consisting of CA125 and age alone demonstrated excellent discrimination and calibration for the identification of ovarian cancer in women presenting to primary care with relevant symptoms. Including additional baseline risk factors, symptom type, and routine blood test results did not materially improve model performance. While the model AUC was only slightly higher than that of CA125 alone, at a fixed sensitivity, Model 1 showed superior specificity and PPV at a range of thresholds. When a ≥1% probability threshold was applied to our cohort, rather than the current CA125 cut-off (≥35 U/mL), 1 in 74 of the additional women identified by the model had ovarian cancer.

### 4.1. Strengths and Limitations

The dataset used in this study was a key strength. Information on candidate variables was identified from a large, routinely collected primary care data source, which is broadly representative of the UK general population, and outcome data were obtained from the NCRAS (the English cancer registry), which is considered the gold standard source for cancer diagnostic information for epidemiological research [13]. However, the use of routinely collected data limited the candidate variables that we could include in our model. For example, family history of ovarian cancer is an established risk factor for the disease but was not included as a candidate variable as it is not routinely recorded in primary care records. Including family history may have introduced bias, as a GP might be more likely to ask about and record family history when there is a strong suspicion of ovarian cancer. 

While CA125 testing is only indicated in women with relevant symptoms in UK primary care, symptoms are not always coded within the GP notes; instead, they may be recorded within the free text which is not available for research purposes [43]. In order to include symptoms as predictor variables, we had to exclude 19,691 women who likely had relevant symptoms that were simply not coded—this reduced sample size. Although CPRD is one of the largest primary care datasets in the world, our sample size was limited by available data on CA125 testing. 

The models developed in this study are intended to aid decision-making in women selected by GPs for CA125 testing and, as such, were developed in an entirely CA125-tested population. It is likely that CA125-tested women are at higher risk of ovarian cancer than women with similar symptoms who were not selected for CA125 testing, so the models may not be generalisable to the non-tested group. 

We internally validated models—applying a cross-validation approach—to assess for model optimism and found that this was minimal. However, external validation in an independent dataset is warranted before the models are used in clinical practice.

### 4.2. Comparison with Existing Literature

In a recent systematic review, we did not identify any other diagnostic prediction models that combined CA125 with symptom variables [10]. Existing primary care diagnostic prediction models, such as QCancer Ovarian [36], were developed in general primary care populations (which included women with and without symptoms) with the aim of identifying higher-risk women for tests such as CA125, whereas the models in this study were developed within an entirely symptomatic CA125 tested population. Similarly, secondary care models such as the Risk of Ovarian Malignancy Algorithm (ROMA) and the Risk of Malignancy Index (RMI) are not comparable as they were developed in populations in which all individuals were known to have a pelvic mass with the aim of distinguishing between benign and malignant masses [44,45]. We have previously reported on the diagnostic accuracy of CA125, as used in the primary care population, and have estimated the probability of ovarian cancer based on patient age and CA125 level [8], but this is the first study to evaluate the diagnostic accuracy of CA125 predicated models within the symptomatic primary care population. 

Baseline patient risk factors, routine blood test results, and the type of symptoms with which patients present have previously been found to be predictive of ovarian cancer in general practice [10] and so were included as candidate variables in this study. However, our results indicate that, in symptomatic women undergoing CA125 testing, the CA125 level is the dominant predictor—the inclusion of other variables, with the exception of age, does not materially improve model performance.

### 4.3. Implications for Research and Practice

In 2015, NICE recommended urgent cancer referral for symptomatic primary care patients when the risk of a particular cancer reached 3% [15]. However, specific guidance on ovarian cancer investigation and referral was not brought into line with this threshold. NICE recommend that symptomatic women with CA125 levels ≥35 U/mL should be referred by their GP for an ultrasound; only if the ultrasound shows evidence of possible cancer do they qualify for an urgent cancer pathway referral. As primary care ultrasound usually takes several weeks to be performed in England [46], this could delay ovarian cancer diagnosis. The model developed in this study could be used to select women for urgent cancer pathway referral, in line with the NICE 3% threshold, thereby helping to ensure that those at higher risk of undiagnosed cancer receive prompt specialist investigation, diagnosis and treatment. 

We recently reported that women with false-negative CA125 results in English primary care took twice as long to be diagnosed with ovarian cancer following testing as women with abnormal CA125 results [20]. Such delays in diagnosis could have a detrimental effect on patient outcomes, including morbidity and survival. Model 1 would allow women with ‘low risk but not no risk’ of ovarian cancer to be identified and offered non-urgent evaluation or interval re-assessment. Applying a ≥1% probability threshold (instead of the current CA125 cut-off), this approach could help detect 1 extra ovarian cancer for every 74 additional patients identified by the model for further evaluation. This evaluation could involve re-testing for CA125, as most women who are retested in primary care prior to diagnosis have rising CA125 levels [20], or a referral for a non-urgent transvaginal ultrasound. An example of a risk-stratified approach using model thresholds is illustrated in Figure 3. As Model 1 is based solely on CA125 level and age, it could readily be incorporated within laboratory computer systems, and the patient’s cancer probability could be reported to the GP alongside the CA125 level.

Implementing a two-tier risk-stratified approach is likely to result in more non-urgent investigation in primary care (‘low risk but not no risk’ women) and more urgent cancer referrals (higher risk women). Any such change in guidelines would require a full health economic evaluation to assess the potential impact of such a strategy on the healthcare service and on patients, as most of those investigated would ultimately not be diagnosed with ovarian cancer. This evaluation would also be valuable in ensuring that the most appropriate model thresholds are chosen for use in clinical practice. The model thresholds evaluated within this study are of particular relevance to the healthcare system in England. However, following validation in appropriate local datasets, our model could be used to select women for further investigation for ovarian cancer in line with any regional or national threshold.

In the current study, we have focussed on optimising the initial testing step within the ovarian cancer diagnostic pathway. However, the timely diagnosis of cancer also depends on the accuracy of subsequent testing steps, most notably ultrasound. Further research is needed to ensure that the use of model-based risk thresholds improves the accuracy of the diagnostic pathway as a whole.

The recently published results from the United Kingdom Collaborative Trial of Ovarian Cancer Screening (UKCTOCS) demonstrated that a multimodal ovarian cancer screening approach did not reduce mortality on long-term follow-up [3]. Given the time required to develop and evaluate new screening approaches, symptomatic detection is likely to remain the principal route for ovarian cancer diagnosis for some years, so optimising the diagnostic pathway for symptomatic women remains important. Despite the earlier stage diagnosis of ovarian cancer in the multimodal screening arm of UKCTOCS, mortality was not reduced. Given this, when evaluating the clinical impact of new approaches that aim to expedite ovarian cancer diagnosis, such as the models presented within this study, survival should be considered alongside other outcomes. 

## 5. Conclusions

A model consisting of age and CA125 level performs well for the detection of ovarian cancer in symptomatic women in English primary care. A risk-based triage system, informed by this model, has the potential to expedite the diagnosis of ovarian cancer in those at high risk of undiagnosed ovarian cancer (through urgent specialist investigation) and ‘low risk but not no risk’ of ovarian cancer (through interval retesting or routine ultrasound). Further research is needed to evaluate the practical impact of implementing such an approach on patients and the healthcare system. 

## Figures and Tables

**Figure 1 cancers-13-02886-f001:**
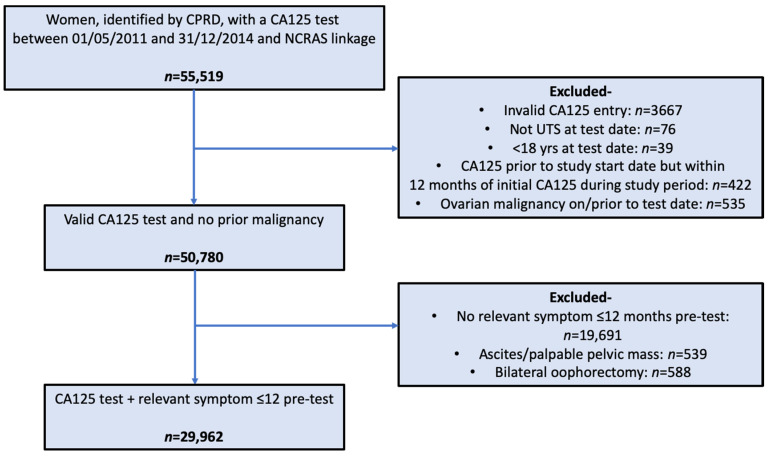
Application of study criteria. “Invalid CA125 entry” = no CA125 value, no or incorrect units, and no upper threshold or spurious upper threshold.

**Figure 2 cancers-13-02886-f002:**
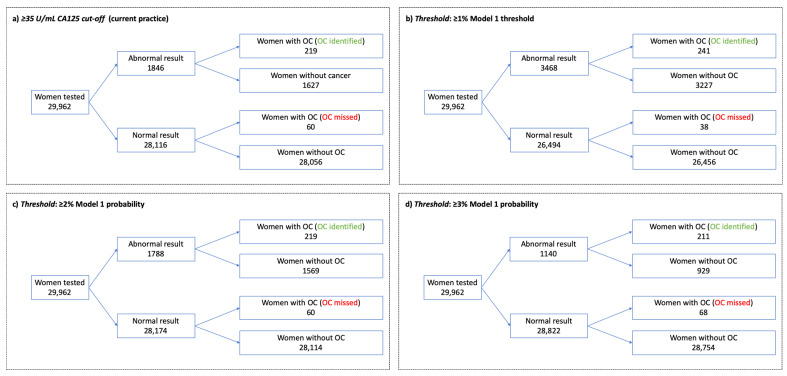
Implications of applying CA125 and model thresholds to the study cohort.

**Figure 3 cancers-13-02886-f003:**
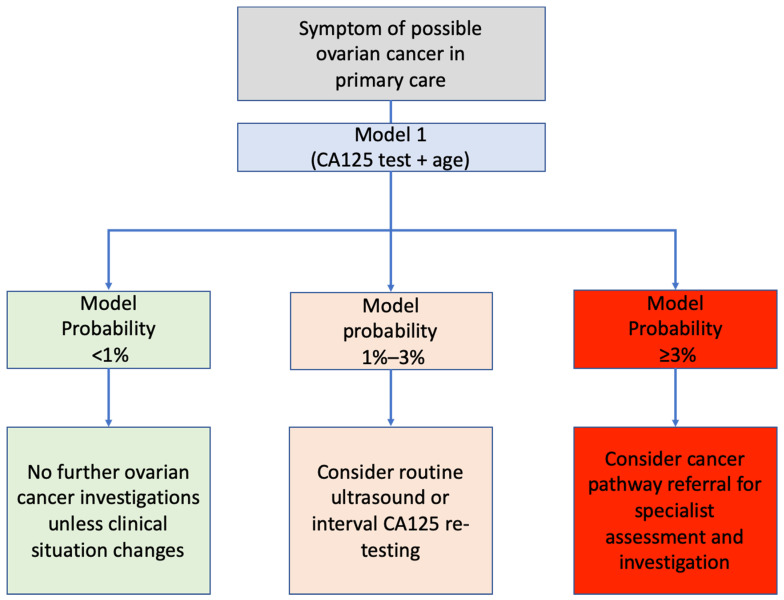
Example risk-based triage system employing model probability thresholds.

**Table 1 cancers-13-02886-t001:** Candidate variables.

Variable	Data Source	Categorisation	Variable Inclusion Time/Period
Risk/protective factors			
Age [26]	CPRD	Continuous (years)	On date of CA125 testing
Ethnicity [25,27]	CPRD and HES	Categorical: White Other ethnicities	Most frequently recorded [24]
Height [28,29,30,31]	CPRD	Continuous (cm)	Most recent on/prior to CA125 test date recorded when ≥18 years old
BMI [28,32,33]	CPRD	Continuous (kg/m^2^)	Most recent on/in the 10 years prior to CA125 test date ≥18 years old
Personal history breast cancer [34]	CPRD/NCRAS	Binary	Up to CA125 test date
Symptoms			
Ovarian cancer symptoms [7]	CPRD	Binary for each symptom. Presence/absence of abdominal/pelvic pain, appetite loss, bloating, distension, change in bowel habit, fatigue, urinary frequency/urgency, new irritable bowel syndrome (≥50 years old), weight loss	12 months prior to CA125 testing
Blood biomarkers			
CA125 [8]	CPRD	Continuous	First valid CA125 level in study period
Albumin [35]	CPRD	Categorical: Not tested <35 g/L ≥35 g/L	Most recent record on or in the 12 months prior to the CA125 test date
Haemoglobin [36]	CPRD	Categorical: Not tested <12 g/dL ≥12 g/d	Most recent record on or in the 12 months prior to the CA125 test date
Platelets [18]	CPRD	Categorical: Not tested <300 × 10^9^/L 300–449 × 10^9^/L ≥450 × 10^9^/L	Most recent record on or in the 12 months prior to the CA125 test date
CRP [17,37]	CPRD	Categorical: Not tested <3 mg/L 3–9.99 mg/L ≥10 mg/L	Most recent record on or in the 12 months prior to the CA125 test date

Abbreviations: BMI = body mass index, CRP = C-reactive protein, CPRD = Clinical Practice Research Datalink, CA125 = cancer antigen 125, NCAS = National Cancer Registration and Analysis Service. Included citations provide further details on the association between candidate variables and cancer risk. The choice of categories for categorical blood tests was based on standard references ranges and the literature.

**Table 2 cancers-13-02886-t002:** Cohort baseline characteristics.

Variable	*n* = 29,962
Risk/protective factors	
Age (years)	Mean = 55 (SD: 15)
Ethnicity:	
White	*n* = 26,511 (88.5%)
Other ethnicities *	*n* = 2217 (7.4%)
Height (cm)	Mean = 162 (SD: 6.8)
BMI (kg/m^2^)	Median = 25.8 (IQR: 22.8–29.7)
Personal history breast cancer	*n* = 1168 (3.9%)
Symptoms	
Abdominal/pelvic pain	*n* = 17,538 (58.5%)
Appetite loss	*n* = 203 (0.7%)
Bloating	*n* = 5649 (18.9%)
Distension	*n* = 821 (2.7%)
CIBH	*n* = 5808 (19.4%)
Fatigue	*n* = 3968 (13.2%)
Urinary frequency/urgency	*n* = 1503 (5%)
≥50 years of age with new IBS	*n* = 286 (1%)
Weight loss	*n* = 1168 (3.9%)
Blood biomarkers	
CA125	Median = 12 (IQR: 8–17)
Albumin:	
Not tested	*n* = 3723 (12.4%)
<35 g/L	*n* = 834 (2.8%)
≥35 g/L	*n* = 25,405 (84.8%)
Haemoglobin:	
Not tested	*n* = 1648 (5.5%)
<12 g/dL	*n* = 3089 (10.3%)
≥12 g/dL	*n* = 25,225 (84.2%)
Platelets:	
Not tested	*n* = 1679 (5.6%)
<300 × 10^9^/L	*n* = 20,442 (68.2%)
300–449 × 10^9^/L	*n* = 7314 (24.4%)
≥450 × 10^9^/L	*n* = 527 (1.8%)
CRP:	
Not tested	*n* = 13,181 (44%)
<3 mg/L	*n* = 6907 (23.1%)
3–9.99 mg/L	*n* = 7370 (24.6%)
≥10 mg/L	*n* = 2504 (8.4%)

* Asian (*n* = 1117), Black (*n* = 562), Mixed (*n* = 160), and Other (*n* = 378). Abbreviations: SD = standard deviation, IQR = interquartile range, IBS = irritable bowel syndrome, CA125 = cancer antigen 125, CRP = C-reactive protein.

**Table 3 cancers-13-02886-t003:** Model discrimination and calibration.

Model	Apparent AUC	Cross-Validation AUC *	Cross-Validation Calibration Slope (95% CI)
Model 1	0.938	0.935	1.01 (0.606–1.42)
Model 2	0.941	0.935	1.05 (0.673–1.42)

* Cross-validation ROC curves are included in Appendix A.

**Table 4 cancers-13-02886-t004:** Diagnostic accuracy metrics for a range of Model 1 thresholds and for CA125 at equivalent sensitivities.

Threshold	Sens	Spec	PPV	NPV
≥1% model probability	86.4 (81.8–90.2)	89.1 (88.8–89.5)	6.9 (6.1–7.8)	99.9 (99.8–99.9)
CA125 of ≥23 U/mL	86.4 (81.8–90.2)	86.0 (85.6–86.4)	5.5 (4.8–6.2)	99.9 (99.8–99.9)
≥2% model probability	78.5 (73.2–83.2)	94.7 (94.5–95.0)	12.2 (10.8–13.9)	99.8 (99.7–99.8)
Ca125 of ≥35 U/mL	78.5 (73.2–83.2)	94.5 (94.3–94.8)	11.9 (10.4–13.4)	99.8 (99.7–99.8)
≥3% model probability	75.6 (70.2–80.5)	96.9 (96.7–97.1)	18.5 (16.3–20.9)	99.8 (99.7–99.8)
CA125 of ≥39 U/mL	75.6 (70.2–80.5)	95.6 (95.3–95.8)	13.8 (12.1–15.7)	99.8 (99.7–99.8)

## Data Availability

This study is based on data provided by the CPRD and the NCRAS and is subject to a licence agreement that prohibits sharing outside the research team. All data are available through CPRD. Data access is subject to approval from the Independent Scientific Advisory Committee for the Medicines and Healthcare products Regulatory Agency.

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
