# Peer review of "Could Ovarian Cancer Prediction Models Improve the Triage of Symptomatic Women in Primary Care? A Modelling Study Using Routinely Collected Data"

_cancers, 2021, doi:10.3390/cancers13122886_

Round 1

Reviewer 1 Report

This is a well conducted study of data in primary care records of women presenting with known symptoms of ovarian cancer who underwent CA125 testing. Data from those diagnosed with ovarian cancer within one year were used to derive models to risk stratify at presentation regarding the likelihood of an ovarian cancer diagnosis and hence expedite diagnosis compared with the current pathway.

The study is well  written and presented and demonstrates that the addition of other routinely measured variables did not improve the  prediction model over and above age and CA125. The authors’ proposal would merit a health economic evaluation to determine viability and prospective evaluation.

Could the authors explain to the reader why they censored ovarian cancer diagnosis at within one year of the blood test.

Author Response

We are grateful to the reviewer for their comments on the manuscript. We agree that an explanation of the rationale for only including ovarian cancers diagnosed in the 12 months following the CA125 test would be useful for the reader and have added this to the methods section (page 3, line 125-131):

“It was assumed that cancer diagnosed within 12 months of the initial CA125 test was present at the time of CA125 testing. It is possible that incidental ovarian cancers might arise and be diagnosed in the 12 months after testing or that it may take more than 12 months following presentation for women to be diagnosed. We chose a period of 12 months, which has been applied widely in previous studies [8,17–20], as a compromise between minimising the inclusion of incidental ovarian cancers and maximising the inclusion of relevant cancers.”

Reviewer 2 Report

This is a well described and presented study on ovarian cancer risk prediction in a symptom-positive setting. I am generally positive towards the article, my only real concerns are to do with the scientific import. 

The authors claim that this is the first study to look at OC prediction in this particular setting (i.e. women with symptoms in primary care) which may be true, I'm not sure, but it is certainly the case that some consider use of symptoms an unhelpful triaging approach with not nearly enough specificity relating to OC. The benefit of symptom-based variables is somewhat controversial as I understand it.

And the results of this study seem to confirm the absolute lack of added predictive ability over and above CA125 and age - which basically then takes you back to the position of screening in a general post-menopausal population... which unfortunately UKCTOCS has now shown to be of no benefit to OC survival even if use of CA125 can produce a stage shift. So, even though the authors argue their position well, I think it would be hard to seriously consider a full scale economic health evaluation for risk prediction and triaging at a symptomatic stage, as they suggest in the discussion. 

Apart from these general misgivings, I felt it was well-written, adhering to Tripod guidelines and had a sound statistical foundation. 

Just a few minor queries and comments below:

Women were selected given a CA125 measurement. I wonder how many otherwise eligible women presenting with symptoms were excluded? I ask because I suspect having a CA125 is somehow indicative of a suspected higher risk rather than simply a chance event. In which case the sample this is based on doesn't really reflect presenting symptomatic women but a higher (or at least differential) risk subset, who in a sense have already been set on that triaging process?

The one real statistical query I have is about the splines used. 5 knots may be the default in some stats packages but it really seems a lot of unnecessary degrees of freedom used up. Looking at the OR CIs in the table there seems to be excessive overfitting of these 2 variables, especially ln(CA125). This seems a particularly strange modelling choice given the EPV<10. In fact the one continuous variable one might a priori suspect a strong non-linear relationship with OC is BMI, which has been left linear (was this checked?) as it has a 'history' of J-shaped dose-response curves with disease outcomes. 

I wonder if instead of cut-offs based on risk probability from the model, set specificity values were considered? That  allows one to control for false-positives which are often deemed the crucial criterion. 

I'm rather amazed the AUC values (especially apparent AUC) were identical to 3dp between the two models. Even though model 2 had 7 'significant' additional variables. This seems incredible?!

It was mentioned a drawback that family history couldn't be included as a risk factor but it got me thinking whether variables like this - the 'baseline risk factors' - should be included at all? Do they not detract (and possibly overwhelm in a modelling sense) from the idea of triaging at the symptomatic stage? These are factors that apply to OC risk before and always, so conceptually I wonder if it is the best idea to include? 

Author Response

Reviewer comment: This is a well described and presented study on ovarian cancer risk prediction in a symptom-positive setting. I am generally positive towards the article, my only real concerns are to do with the scientific import. 

The authors claim that this is the first study to look at OC prediction in this particular setting (i.e. women with symptoms in primary care) which may be true, I'm not sure, but it is certainly the case that some consider use of symptoms an unhelpful triaging approach with not nearly enough specificity relating to OC. The benefit of symptom-based variables is somewhat controversial as I understand it.

And the results of this study seem to confirm the absolute lack of added predictive ability over and above CA125 and age - which basically then takes you back to the position of screening in a general post-menopausal population... which unfortunately UKCTOCS has now shown to be of no benefit to OC survival even if use of CA125 can produce a stage shift. So, even though the authors argue their position well, I think it would be hard to seriously consider a full scale economic health evaluation for risk prediction and triaging at a symptomatic stage, as they suggest in the discussion. 

Author reply: We are very grateful to the reviewer for their feedback on the manuscript.

Many countries (including the UK where this study was performed) currently recommend that doctors consider performing tests for ovarian cancer (e.g. CA125), in women presenting with relevant symptoms in primary care [1]. Most women with ovarian cancer are currently diagnosed after presenting to their doctor with a relevant symptom [2,3]. As UKCTOCS did not demonstrate a long-term survival benefit, detection following symptomatic presentation is likely to remain the principal route to diagnosis for women with ovarian cancer for the foreseeable future. Given this, we believe it is important that we optimise the current diagnostic pathway for symptomatic women.

It is important to note that all women in this study had a possible symptom of ovarian cancer prior to CA125 testing – the incidence of ovarian cancer in our cohort was much higher than in screening studies despite the fact that we included younger women [4]. Previous studies have quantified the predictive ability of individual symptoms in primary care, demonstrating that individual symptoms differ in their positive predictive values [5]. We included presenting symptoms in Model 2 to account for this. We have added a line highlighting this to the methods (page 3, line 140-143):

“While all patients in this study had a symptom of possible ovarian cancer prior to CA125 testing, previous research has demonstrated that these symptoms differ markedly in their predictive value [19]. Symptoms were therefore included as variables to account for this.”

Reviewer comment: Apart from these general misgivings, I felt it was well-written, adhering to Tripod guidelines and had a sound statistical foundation. 

Just a few minor queries and comments below:

Women were selected given a CA125 measurement. I wonder how many otherwise eligible women presenting with symptoms were excluded? I ask because I suspect having a CA125 is somehow indicative of a suspected higher risk rather than simply a chance event. In which case the sample this is based on doesn't really reflect presenting symptomatic women but a higher (or at least differential) risk subset, who in a sense have already been set on that triaging process?

Author reply: Thank you for highlighting this. We agree that women selected for CA125 testing are likely to be at higher risk than those with symptoms not selected for CA125 testing. Our models are specifically intended for use in women who have been selected by GPs for CA125 testing in order to improve the triage of these women. We have added a line to the strengths and limitations section highlighting that our models may not be generalisable to women not selected for CA125 testing (page 10, line 292-297):

“The models developed in this study are intended to aid decision making in women selected by GPs for CA125 testing and, as such, were developed in an entirely CA125-tested population. It is likely that CA125-tested women are at higher risk of ovarian cancer than women with similar symptoms who were not selected for CA125 testing and so the models may not be generalisable to the non-tested group.”

Reviewer comment: The one real statistical query I have is about the splines used. 5 knots may be the default in some stats packages but it really seems a lot of unnecessary degrees of freedom used up. Looking at the OR CIs in the table there seems to be excessive overfitting of these 2 variables, especially ln(CA125). This seems a particularly strange modelling choice given the EPV<10. In fact the one continuous variable one might a priori suspect a strong non-linear relationship with OC is BMI, which has been left linear (was this checked?) as it has a 'history' of J-shaped dose-response curves with disease outcomes. 

Author reply: We used 5 knots for age and CA125 based on the results of an earlier study [6], in which Akaike Information Criterion (AIC) was employed to select the most appropriate number of knots in line with Harrells recommendations. We have added a line to the methods to highlight this (page 6, line 173-175):

“ The choice of 5-knots was pre-specified based on the results of previous research which used the study data to explore the relationship between CA125 level, age and ovarian cancer [8].”

As the cross-validation AUC was similar to the apparent performance AUC for Model 1 (which only contained age and CA125 level) this decision does not appear to have contributed to significant model overfitting.

In order to limit the degrees of freedom for Model 2, we only included splines for the two continuous variables (CA125 level and age) which we have previously shown affect ovarian cancer risk in CA125 tested women and which were prespecified in this study. The reviewers comment prompted us, post-hoc, to explore the relationship between ovarian cancer and BMI by including polynomial terms in a logistic regression model alongside log BMI. We found that the p value of log BMI2 was not significant indicating that, in this instance, assuming a linear association was appropriate.

Reviewer comment: I wonder if instead of cut-offs based on risk probability from the model, set specificity values were considered? That  allows one to control for false-positives which are often deemed the crucial criterion. 

Author reply: We chose to focus on cancer probability in order to align our study findings with United Kingdom guidelines which use cancer probability when making recommendations e.g. the UK guideline producing body NICE recommend urgent cancer pathway referral if the probability of cancer is ≥3%. We agree with the reviewer that specificity is also important and have provided specificities for the model thresholds in Table 4.   

Reviewer comment: I'm rather amazed the AUC values (especially apparent AUC) were identical to 3dp between the two models. Even though model 2 had 7 'significant' additional variables. This seems incredible?!

Author reply: Many thanks for highlighting this. We reviewed the results and noted a transcription error – the apparent performance of Model 2 presented in Table 3 should be 0.941 rather than 0.938. Apologies for this error – we have corrected the table. The cross-validation AUCs are presented correctly in the table and the interpretation of the results remains the same.

Reviewer comment: It was mentioned a drawback that family history couldn't be included as a risk factor but it got me thinking whether variables like this - the 'baseline risk factors' - should be included at all? Do they not detract (and possibly overwhelm in a modelling sense) from the idea of triaging at the symptomatic stage? These are factors that apply to OC risk before and always, so conceptually I wonder if it is the best idea to include?

Author response: Thanks for highlighting this interesting methodological point. We chose to include baseline risk factors for two reasons. Firstly, we have previously found that the baseline risk factor of age markedly affects the probability of ovarian cancer in CA125 tested women [6]. Secondly, other models used to identify patients at risk of undiagnosed cancer, including the widely used QCancer models [7], include a variety of baseline risk factors alongside symptoms and test results. These studies have found that the inclusion of such variables can improve model performance for various cancers. While In our study the inclusion of baseline risk factors, other than age, did not improve model performance it was not possible to know this prior to modelling.

Reference

[1] Funston, G.; Melle, V.M.; Ladegaard Baun, M.-L.; Jensen, H.; Helpser, C.; Emery, J.; Crosbie, E.; Thompson, M.; Hamilton, W.; Walter, F.M. Variation in the initial assessment and investigation for ovarian cancer in symptomatic women: a systematic review of international guidelines. BMC Cancer 2019, 19, 1028

[2] Barrett, J.; Sharp, D.J.; Stapley, S.; Stabb, C.; Hamilton, W. Pathways to the diagnosis of ovarian cancer in the UK: A cohort study in primary care. BJOG 2010, 117, 610–4.

[3] National Cancer Intellegence Network. Routes to Diagnosis 2006-2016 by year, V2.1a Available online: http://www.ncin.org.uk/publications/routes_to_diagnosis (accessed on May 21, 2020).

[4] Gilbert, L.; Basso, O.; Sampalis, J.; Karp, I.; Martins, C.; Feng, J.; Piedimonte, S.; Quintal, L.; Ramanakumar, A. V.; Takefman, J.; et al. Assessment of symptomatic women for early diagnosis of ovarian cancer: results from the prospective DOvE pilot project. Lancet Oncol. 2012, 13, 285–291.

[5] Hamilton, W.; Peters, T.J.; Bankhead, C.; Sharp, D. Risk of ovarian cancer in women with symptoms in primary care: population based case-control study. BMJ 2009, 339, b2998.

[6] Funston, G.; Hamilton, W.; Abel, G.; Crosbie, E.J.; Rous, B.; Walter, F.M. The diagnostic performance of CA125 for the detection of ovarian and non-ovarian cancer in primary care: a population-based cohort study. PLoS Med. 2020, 17, e1003295.

[7] Hippisley-Cox, J.; Copeland, C. QCancer Tool Available online: https://qcancer.org (accessed on Aug 9, 2020).